# Using the Women Empowerment in Livestock Index (WELI) to Examine Linkages between Women Smallholder Livestock Farmers’ Empowerment and Access to Livestock Vaccines in Machakos County of Kenya: Insights and Critiques

**DOI:** 10.3390/vaccines10111868

**Published:** 2022-11-04

**Authors:** Catherine Kaluwa, Jemimah Oduma, Faduma Abdullahi Abdirahman, Byalungwa Kyotos Kitoga, Angela A. Opondoh, John Muchibi, Brigitte Bagnol, Marieke Rosenbaum, Sylvia Onchaga, Meghan Stanley, Janetrix Hellen Amuguni

**Affiliations:** 1Department of Veterinary Anatomy and Physiology, University of Nairobi, Nairobi P.O. Box 30197-00100, Kenya; 2Department of Animal Production, University of Nairobi, Nairobi P.O. Box 29053-00625, Kenya; 3Institute of Anthropology, Gender, and African Studies, University of Nairobi, Nairobi P.O. Box 30197-00100, Kenya; 4Fairdeal Agrivet and Services Ltd., Nairobi P.O. Box 475-00521, Kenya; 5Cummings School of Veterinary Medicine, Tufts University, 200 Westboro Road, North Grafton, MA 01536, USA

**Keywords:** Women’s Empowerment in Livestock Index Tool, WELI, agency, Kenya, female livestock keepers, livestock vaccines, Newcastle Disease, Contagious Caprine Pleuropneumonia

## Abstract

Livestock diseases are a major barrier to productivity for both male and female livestock keepers in Africa. In Kenya, two of the most devastating livestock diseases are Newcastle Disease (ND) in poultry and Contagious Caprine Pleuropneumonia (CCPP) in goats. Female livestock keepers tend to own more small ruminants (goats, sheep, etc.) and poultry and their livelihoods are adversely affected if their herds are not vaccinated against these diseases. Livestock farming has gender specific challenges and opportunities, with implications for the empowerment of women smallholder farmers, their household well-being, food security, and livelihoods. There is a need to estimate the level to which women benefit personally, socially, and economically from keeping livestock, yet there are very few studies that can measure if livestock production does in fact empower women smallholder livestock farmers. This study was done to examine linkages between women’s empowerment and access and control over livestock products and vaccines. The Women Empowerment in Livestock Index (WELI) tool, which was customized to include questions on livestock vaccine access, was used to capture baseline data on empowerment scores for women in Machakos county, Kenya, prior to implementation of animal health and vaccine test models. In total, 400 participants were surveyed in two wards of Machakos County, Kola and Kalama, which were purposively selected. Women’s empowerment was mapped to three domains (3DE): intrinsic agency (power within), instrumental agency (power to), and collective agency (power with) measured against adequacy in 13 indicators. Our results indicate that the household structure (female headed or dual headed household), age of respondents and number of members in a household influence the adequacy score. Work balance was the most significant negative contributor to women’s disempowerment. Women contributed the most to livestock productive activities and attained adequacy in this area compared to men, directly impacting the WELI score. Women smallholder livestock farmers report low CCPP and ND vaccination rates, minimal knowledge on livestock diseases, a lack of access to cold chain storage and rarely visited veterinarians. The WELI score was 0.81 indicating a high level of empowerment for women in this community compared to men leading us to conclude that the overall WELI score was not an accurate indicator of women‘s empowerment in Machakos County. However, the decomposability of the index allows us to disaggregate the drivers of change and to examine how individual indicators contribute to disempowerment.

## 1. Introduction

Livestock diseases are a major barrier to productivity for both male and female livestock keepers around the world. These diseases are damaging both at a national level, costing developing countries millions of dollars in lost revenue, as well as at an individual level, limiting the productivity of individual livestock keepers [1,2]. In Kenya, a country that derives 12% of its national Gross Domestic Product (GDP) from the livestock sector, two of the most devastating and influential livestock diseases experienced by livestock keepers are Newcastle Disease (ND) in poultry and Contagious Caprine Pleuropneumonia (CCPP) in goats [3]. The poultry industry accounts for approximately 30% of the agricultural sector and in 2015 contributed 7.8% of the national GDP [4]. Endemic Newcastle disease (ND) is a major constraint for increased productivity due to frequent outbreaks despite intensive vaccinations [5]. Goat keepers, on the other hand, are faced with Contagious Caprine Pleuropneumonia (CCPP), a highly contagious respiratory disease with a mortality rate of 70% and a morbidity rate ranging from 80 to 100% [6]. Annual economic losses in Kenya due to CCPP for a standard herd of 100 head have been estimated in some models to be as high as €1712.66 [6]. An internationally reportable disease, CCPP is rarer than ND but equally important for Kenyan livestock keepers. Vaccination against these diseases is essential to reduce outbreaks, improve livestock productivity and enhance animal sourced food quality, and ultimately contribute to sustainable livelihoods [7].

Women play a central role in most countries as food producers and providers [8] and control (some) livestock products that are essential for food and nutrition security [9]. Women constitute 70% of food producers and providers in Kenya and represent the majority of poor livestock keepers [10]. Raising livestock, as opposed to crops, tends to be a more accessible agricultural pursuit for women and as a result they rely on their animals more heavily than their male counterparts [11]. Studies have shown that female livestock keepers tend to own more small ruminants (goats, sheep, etc.) and poultry than large livestock (water buffalo, cows, etc.) [12,13]. Livestock farming has gender specific challenges and opportunities, with specific implications for the empowerment of women smallholder farmers, their household well-being, food security, and livelihoods [14]. As a result of this differential ownership, women may be more affected by small ruminant and poultry diseases, such as ND and CCPP. There are also a number of general barriers and constraints that female livestock keepers face that may exacerbate the effects of these diseases on their livelihood. Women have limited access to services, credit, technology, training and information regarding livestock, putting them at greater risk of livestock loss [9]. Time lost to unpaid care work and other time costs [10,15] may limit the time available for women to tend their livestock. All of these factors put female livestock keepers at a higher risk of losing animals to ND and CCPP.

There is a need to estimate the level to which women benefit personally, socially and economically from keeping livestock and being engaged in livestock projects. Although some figures estimate the limited agricultural productivity of women to be approximately 20–30% less than their male counterparts [15], there are very few studies that can measure if livestock production does in fact empower women smallholder livestock farmers. Championing the empowerment of smallholder livestock farmers enhances food security and contributes to gender equality, access and control over resources, while giving women agency [9,16].

Women’s empowerment is an indicator of women’s agency and enhances their ability to improve their decision making and attain instrumental outcomes, such as improved household well-being (increased productivity, food security and livelihoods) [17,18] and their greater control over their resources and their bodies [19]. Thus, the measurement of women’s empowerment is a strategic component for evidence-based development policy. Indeed, the 2030 Agenda for Sustainable Development prioritizes women’s empowerment in Sustainable Development Goal (SDG) #5: to achieve gender equality and empowerment among all women and girls [20].

Despite a growing commitment to gender equality and women’s empowerment and the proliferation of women’s empowerment measures, consistent approaches for measuring women’s empowerment in livestock development projects are lacking. Tools to determine women’s empowerment aim to describe the capability of women for self-determination: to take control over their own circumstances and to realize their aspirations in order to live a life they have reason to value [21,22,23]. Many livestock development interventions and projects aim to empower women alongside goals to improve livestock productivity and income; increase vaccine accessibility; reduce poverty, hunger, and undernutrition; and improve health outcomes. In spite of this increasing commitment to gender equality and women’s empowerment and the increase in the number of women’s empowerment tools and measures, reliable approaches for measuring women’s empowerment in livestock development projects are lacking. Appropriate metrics are needed to assess whether these projects are achieving their goals. The same applies to the agricultural sector. There are few indices that depict control over resources or agency within the agricultural sector in which women account for 43% of the agricultural labor force in developing countries [10]. Oxfam’s Women’s Empowerment Index framework, for example, provides a numerical value for empowerment, while at the same time instituting causation when incorporated within impact evaluation schemes [24]. However, there is a lack of consensus on what domains constitute women’s empowerment and how to measure women’s empowerment across different sectors and countries. Indices, such as the Gender-related Development Index (GDI) and Gender Empowerment Measure (GEM), highlight economic and educational aspects of women’s empowerment and gender equality [25] and the Women, Peace and Security Index focuses on social inclusion, justice and security [26]. Yet, these measures overlook significant domains of women’s empowerment, such as women’s self-reported human, social and economic resources for empowerment [22], as well as attitudinal and behavioral evidence of empowerment, such as women’s attitudes about gender and violence against women, their freedom of movement and their domestic, sexual, and reproductive decision-making [27,28]. Similarly, the Gender Gap Index report [29], the Gender Development Index (GDI), and the Gender Inequality Index (GII) that were/are reported by the United Nations Development Programme (UNDP) Human Development Reports cover gender inequalities in a broad set of domains but do not measure empowerment directly [30]. Longwe’s Women’s Empowerment Framework constituted five elements: welfare, access, conscientization, participation and control, which has been used to evaluate women’s empowerment projects, is limited because it does not include the cognitive elements of conscientization and self-esteem, and therefore is not multi-dimensional [31]. A notable methodological weakness is that these, as well as the Gender Empowerment Measure (GEM), all use aggregate data and hence cannot be decomposed by age, region, or other social groups.

The Women’s Empowerment in Agriculture Index (WEAI) was created in 2012 initially as a tool to monitor women’s empowerment from the US government’s Feed the Future Initiative. It is a survey-based index designed to measure the empowerment, agency, and inclusion of women in the agricultural sector through a focus on women’s agency. It uses individual-level data collected from male and female household members in a household survey designed for this purpose [32]. Despite its reliability in certain agricultural contexts, the Pro-WEAI cannot be successfully applied in settings where livestock farming is the dominant form of livelihood and requires adaptation. Livestock farming contributes 33% of GDP globally [14,33]. Yet, only 30% of questions in the WEAI focus on livestock. Therefore, using the PRO-WEAI as a starting point, a team of researchers at the International Livestock Research Institute (ILRI) and Emory University developed the Women’s Empowerment in Livestock Index (WELI), a new index to assess the empowerment of women in the livestock sector. WELI explores how livestock is related to and supports women’s empowerment and the health and nutrition of women and children [14]. The WELI focuses on key areas of livestock production, such as animal health, breeding, and feeding, as well as the use of livestock products, such as animal-source-food processing and marketing. The WELI uses a series of designed questions to calculate a single score representative of the level of empowerment of an individual woman or man within the livestock sector. WELI uses the same 3DE-as Pro-WEAI aligned with 13 indicators: 12 that are the same as Pro-WEAI and one extra indicator that is focused on input in livestock productive decisions. The WELI survey includes an extra module focused on livestock roles, access and control over livestock related resources.

For this study, the WELI survey was used to capture baseline data on empowerment scores for women in Machakos County prior to implementation of animal health and vaccine test models. This would allow us to examine the linkages between women’s empowerment and access and control over livestock products and vaccines, identify which indicators and dimensions of women’s empowerment are related to access and control over livestock, livestock products and vaccines as well as factors that contribute the most to disempowerment of women in Machakos County so as to target specific interventions that can improve access to and adoption of livestock vaccines. We also test whether women’s empowerment has differential associations for dual headed and single headed households and households with different numbers of children. A project specific modification was made to the WELI tool to include an additional module pertaining to vaccination and various livestock diseases including CCPP and ND to capture the relationship between livestock vaccination and empowerment.

## 2. Materials and Methods

### 2.1. Study Area

The study was conducted in Machakos Town sub-county, Kenya, which is located 61.6 km southeast of Nairobi, Kenya’s capital city. Machakos Town sub-county has 7 wards, from which Kola and Kalama wards were purposively selected because they own chickens and goats (Figure 1). Machakos Town Sub-county’s population is estimated to be 170,606. The climate is semi-arid, and the county has an altitude of 1000 to 2100 m above sea level. It lies between latitudes 0.45′ S and 1.31′ S and longitudes 36.45′ E and 37.45′ E and covers an area of 6850 km^2^. The average rainfall ranges from 500–1300 mm, and the average temperature is 18–25 °C. Subsistence agriculture is the main farm activity. Maize and drought-resistant crops, such as sorghum and millet, are grown due to the area’s semi-arid state.

### 2.2. Sampling and Data Collection

The Women’s Empowerment in Livestock Index (WELI), a standardized quantitative survey tool to measure empowerment of women in the livestock sector at the level of the household and along the Vaccine Value Chain (VVC) was used. WELI, was customized for the project to include questions on Vaccines. The survey was administered using the Open Data Kit collect tool (ODK), which is an android app that is used in survey-based data collection. This app was uploaded onto tablets and phones. A three-day training on the WELI was conducted prior to fieldwork to provide enumerators with skills and practice to conduct the survey using ODK. The country team and students attended the training at ILRI offered by experts from ILRI. The WELI was then administered to a sample of 300 female local farmers and 100 male members of households at their farms or homes. Additionally, focus group discussions (FGDs), key informant interviews (KIIs) and stakeholder meetings were done to obtain qualitative information on how the communities define empowerment (Table 1).

The field application of the WELI was conducted between 21 October and 30 November 2019 in both wards of Kola with 49 villages and Kalama with 98 villages that make up the Kenya study site. The wards are further divided into location and sub locations. All the 49 villages in Kola and 98 villages in Kalama were subjected to a computer generated randomization (each ward separately as they were surveyed at different times) to generate a random list of 1–49 in Kola and 1–98 in Kalama. Since Kalama is twice the size of Kola, it was purposely decided that the number of villages and ultimately number of households surveyed would reflect this and therefore the numbers in Kola would be half the numbers in Kalama. Out of the random lists, a number of villages were serially picked to cover at least 57% of the ward. Therefore in Kola, 27 villages were surveyed while in Kalama, 56 villages were surveyed. From the selected village’s households (HH) to be surveyed were randomly selected again using the random walk system. Care was taken to make sure that each location or sub location was represented by at least one village. The number of HH from the selected villages was proportional to the size of the village and overall wards. Therefore in Kola, 100 HH with women (25 interviews with men from the same household) were surveyed compared to 200 (75 with men) in Kalama, giving a total of 300 HH; 300 females and 100 males for a total of 400 surveys. A HH coding and tracking system was designed and used by the enumerators. The sample size of 400 was based on sample size calculations provided by developers (ILRI) of the WELI tool on what minimum effective sample size would be [14]. For the data analysis, a CSV file was created using the R statistical tool, and the data was analyzed using Stata 16.0 for Windows (StataCorp LLC, College Station, TX, USA). Analysis “do files” were provided by ILRI.

### 2.3. Focus Groups/Stakeholder Meetings on Local Understanding of Empowerment

Prior to the quantitative data collection, FGDs, KIIs and stakeholder meetings were held to discuss the local understanding of the term empowerment. To ensure proper translation, participants were asked to describe their most desired outcome or vision, or best case scenario/or highest achievable goals, or the point at which they considered themselves most successful as the ultimate empowerment description. Exercises done included asking participants to ‘Imagine what their ideal/dream life (or a life they would like for themselves) would look like 10 years from now and what needed to happen to them in order to make their ideal/dream life come true?’ Discussions provided comments, such as, e.g., ‘I need to be…, I need to get…’; ‘My community needs to allow me to....’. This idea of ‘what needs to happen to them to realize their ideal life’ as a concept of empowerment was used to discuss the characteristics of an empowered man/woman/chicken woman farmer and or goat woman farmer.

### 2.4. WELI as a Quantitative Tool

WELI is a population level aggregate index, reported at the country or sub-national level and composed of two sub-indices. The first sub-index assesses the degree to which respondents are empowered in three domains of empowerment (3DE): intrinsic agency (power within), instrumental agency (power to), and collective agency (power with). A total of 13 indicators are aligned with these 3 domains [14,34]. The WELI is very closely adapted from Pro-WEAI with similar domains (3DE) and uses the same 12 Pro-WEAI indicators plus 1, the livestock productive decisions, for a total of 13 indicators. The four indicators of intrinsic agency include autonomy in income, self-efficacy, attitudes about IPV against women, and respect among household members. The seven indicators of instrumental agency include input into productive decisions, input into livestock productive decisions, ownership of land and other assets, control over use of income, access to and decisions on financial services, work balance, and visiting important locations. Collective agency is comprised of group membership and membership in influential groups. The 3DE reflects the percentage of women and men who are empowered and, among those who are not, the percentage of domains in which they achieve a pre-defined threshold for adequacy in empowerment [34]. The second sub-index, the GPI, measures gender parity. The GPI reflects the percentage of women who are empowered or whose achievements are at least as high as the males in their households. For those households that have not achieved gender parity, the GPI shows the empowerment gap that needs to be closed for women to reach the same level of empowerment as men in their households [32]. The weights of the 3DE and GPI sub-indices are 90% and 10%, respectively. The choice of weights for the two sub-indices follows the original Women’s Empowerment in Agricultural Index-WEAI [34], placing greater emphasis on the 3DE, while still recognizing the importance of gender equality as an aspect of empowerment and also reflecting the different magnitudes of the indices. The total WELI score is the weighted sum of the 3DE and GPI. Improvements in either 3DE or GPI will increase WELI scores. The computation of the WELI follows the original WEAI computation as reported in [32] and the Pro-WEAI [34] and is published in the two journal articles [14,34]. Data collection for the WELI focuses on the two livestock species within that community that are most relevant for women’s empowerment: these are the species that the respondent considers to be most important for household well-being as well as the woman’s well-being. For purposes of this study, these were goats and chicken. The formative process in developing WELI is described in detail by [14].

### 2.5. Computation of the WELI Score

The respondent’s overall empowerment score was computed as the weighted average of her/his adequacy scores in the 13 indicators (all weighted 1/13). If they met adequacy in 9 out of the 13 indicators, then that individual was classified as empowered. Conversely, if they had not met adequacy in 9 out of the 13 indicators, then that individual was considered disempowered. The individual level scores were aggregated to construct WELI (weighted mean of two sub-indices): the three Domains of Empowerment Index (3DE), with a weight of 90 percent and the Gender Parity Index (GPI), with a weight of 10 percent [34].

The GPI is constituted by households where two adults, the man and woman were interviewed. A total of 78 households with complete data for the 2 household members were used to construct the GPI. A data management plan (DMAP) was created for the WELI data analysis.

### 2.6. Additional Livestock Vaccine Module

Since WELI only focuses on livestock production, our study chose to add an additional module focused on livestock vaccines to the original WELI to capture the relationship between livestock vaccination and empowerment. This module was analyzed separately as indicated in the Table 2 below.

### 2.7. Data Analysis

Data was analyzed using Stata version 15 for Windows (StataCorp LLC, College Station, TX, USA). The analysis focused on describing demographic characteristics of the respondents, their adequacy across the 13 indicators, empowerment level (3DE and overall WELI score), factors contributing to disempowerment and participation in livestock farming activities by gender. Specifically, on vaccination, the relationship between empowerment and (i) access to livestock vaccine information, (ii) ability to vaccinate their livestock against ND or CCPP, and (iii) number of animals lost to ND or CCPP in the past one year, was evaluated using regression analysis.

### 2.8. Ethical Approval

Ethical approval for human subjects’ research was obtained locally in Kenya (country clearance via National Commission for Science, Technology & Innovation #NACOSTI/P/19/80106/28666; ethical approval via University of Nairobi Faculty of Veterinary Medicine Biosafety, Animal Use and Ethics Committee #FVM BAUES/2019/194) and through the Tufts University Social Behavioral & Educational Research Institutional Review Board (#1907033) prior to commencement of research activities. A standardized written informed consent document was used to obtain consent from participants prior to KIIs, FGDs, WELI, and stakeholder meetings.

## 3. Results

The results of the study have been organized into three subsections. Section 3.1 presents the descriptive data including the socio-demographic data, participation of respondents in livestock production and agricultural activities and relationship to empowerment. Section 3.2 presents the 3DE and WELI indicators and includes the community’s definition of empowerment, the three domains of empowerment that encompass the 13 WELI indicators, factors that contribute to disempowerment, absolute WELI score, and the relationship between the 3DE scores and demographic data. Section 3.3 presents the data on access to knowledge and vaccination against CCPP and ND, and relation between empowerment and vaccination status.

### 3.1. Participant Descriptive Information

#### 3.1.1. Socio-Demographic Profiles

The socio-demographic profile of respondents provides context on the study population. Of the 400 respondents interviewed, 381 respondents had complete interviews/data (95.3%). Of these, 75.1% were female and 24.9% male. In total, 248 women (86.7%) were from dual male-female adult households, 38 (13.3%) were from female-only adult households, and 95 were from male-only households. Male interviewers were any male members of households: husbands, sons, and brothers. Reasons for non-response/missing interviews were (i) incomplete data (ii) missing or mis-matched male/female HH information (iii) failure to consent.

The female and male participant age ranged from 22–85 years with a median of 48 years and 25–90 years with a median of 59 years, respectively. There was a significant difference in the average age between the female and male participants (Two-sample Wilcoxon rank-sum test z = 4.422, *p*-value < 0.001). Table 3 below provides the socio-demographic data of participants).

In the sampled households, the average household has five members, one of which is a child aged (0–18 years). The average number of children living in a household is 2 (median = 2, IQR: 1–3 children). Figure 2 shows the distribution of household size and number of children living in the surveyed households, respectively. Household size differs significantly by house type (*p*-value < 0.001); a majority (74%) of the female adult only households have 4 members and below. At least one-third (37%) of female-adult only households have no children compared to 21% of dual male and female adult households.

#### 3.1.2. Relationship between WELI Score and Age/Household Size

We run a generalized linear regression model using the continuous WELI score as the dependent variable to test the hypothesis that men have higher empowerment scores than women and that older women or men have higher empowerment scores than younger women or men. The results (Table 4) show that there exists a statistically significant relationship between the respondent’s age and WELI scores, after adjusting for gender. This coefficient was significant at a 5% level. We find that for each year’s increase in the respondent’s age, the 3DE score increased by a factor of 0.3%, and for each additional household member, the 3DE score decreased by a factor of 1.9%.

#### 3.1.3. Participation in Livestock Activities

The aim of this section is to determine whether empowered women feel that they can participate to a great extent in vaccinating their animals or if their perceived extent of involvement is inversely related or not at all. We begin by exploring the participation of women and men in important livestock activities.

We found a significantly higher proportion of women than men participating in 7 out the 10 important livestock activities (Table 5).

#### 3.1.4. Comparison of Agricultural and Livestock Activities Verses Empowerment Level

We examined the relationship between the number of agricultural activities and livestock activities a respondent is involved in versus empowerment level. This was done by running a multiple generalized linear regression model with 3DE score as the dependent variable, against the two covariates, and adjusting for gender, age, and household size. We observe (Table 6) a positive and statistically significant relationship between the number of activities a farmer is involved in and the 3DE score at a 5% significance level. Holding other factors constant, for each unit increase in the number of agricultural and livestock activities an individual is involved in, the expected 3DE score increased by a factor of 7.0% and 2.0%, respectively. We also notice an inverse and statistically significant relationship between the household size and 3DE score. For each additional member in the household, an individual’s expected 3DE score decreases by a factor of 1.9%, holding other factors constant.

#### 3.1.5. Participation and Access to Vaccines/Preventative Care and Information

Participants answering questions regarding poultry were mostly female (74%) with a mean age of 51 years old and a range of ages between 22 to 90 years old. The adjusted sample size for the poultry analysis was 152 after data cleaning. The majority (76%) of respondents felt that they had input into most or all decisions regarding their animals, did not vaccinate against ND (~60%), felt that they had a small extent of knowledge regarding poultry health (~62% for women), did not have access to cold-chain for vaccine storage (91%) and jointly owned their poultry (77%). The vast majority of participants reported never or rarely going to a veterinarian (90%), that they are treated with respect by their spouses (82%) and feel self-confident (79%). Overall, the mean number of animals reportedly lost to ND in the last 12 months was substantial at approximately 5.5 animals per household. Participants answering questions regarding small ruminants (goats) were mostly female as well (71%) with a mean age of 51 years and a range of ages between 22 to 90 years old. The adjusted sample size for the goat analysis was 184 after data cleaning. Approximately half of the respondents felt that they had input into most or all decisions (53%), the majority did not vaccinate against CCPP (~90%), felt that they had a small extent of knowledge regarding goat health (73%), did not have access to cold-chain for vaccine storage (97%), and jointly owned their goats (81%). The vast majority of participants reported never or rarely going to a veterinarian (91%), that they are treated with respect by their spouses (77%) and feel self-confident (85%) (Figure 3). Overall, the mean number of animals reportedly lost to CCPP was low at approximately 0.6 animals/household.

### 3.2. Definition of Empowerment, WELI Indicators, and Related Empowerment Scores

#### 3.2.1. Participation and Access to Vaccines/Preventative Care and Information

Participants described empowerment in terms of their ability to access or control resources or make decisions. “Empowerment is doing something that will help me in future. By future I mean from your program I understand that the chicken I rear can help me in future. And also, my husband cannot demand to know the reason I am selling chicken or goats”.

“*I feel empowered because when I don’t have cash and I have my poultry I can sell them. If I sell 10 that is 10,000 at least I can settle some of the bills I have. So empowerment comes from the income I get from them*”. The main areas continuously mentioned were economic independence, increased knowledge and skills on vaccination, increased access to resources such as owning more chicken and goats, more networking and ability to influence household decision making. Participants generated many ideas on what was needed to improve women’s empowerment. Some participants struggled with the concept because they felt it was a status they had never visualized.

“*I have never been empowered by anyone; I just have the knowledge. I don’t want to be idle; I have to work so that I don’t borrow anything especially from the merry-go-around.*” Through focus group discussions, participants generated ideas on what they considered were characteristics of empowered women in Table 7 below. These were grouped into four main buckets; economic independence, knowledge and skills, opportunities for networking, and autonomy in decision making.

#### 3.2.2. Women Empowerment Domains

We describe the empowerment level of women in this study using three domains, namely intrinsic agency, instrumental agency, and collective agency. As reported earlier, the combined indicators in these domains add up to the 13 WELI indicators. Each respondent is classified as either adequate (=1) or inadequate (=0) in a given indicator by comparing their responses to the survey questions with a given threshold.

Intrinsic Agency—this refers to the ‘power within’, that is the process by which one develops a critical consciousness of their own aspirations, capabilities, and rights. In this study the respondent’s intrinsic agency is assessed using four indicators: autonomy in the use of income from agricultural and non-agricultural activities, self-efficacy, their attitudes about domestic violence, and respect among household members. The study found (Table 8) that overall, at least half of the respondents had attained adequacy in the four indicators. We find that the performance of women and men varies by indicator, while the percentage of men achieving adequacy in terms of autonomy in income use (*p*-value = 0.019) and respect among household members (*p*-value = 0.041) was significantly higher compared to women at a 5% level of significance; there was no difference in self-efficacy and attitudes about domestic violence indicators. There was a notable difference between dual female households and female only households in autonomy in income (79/66.9) and respect among households (65.8/47.6).

Instrumental agency—also known as ‘power to’, refers to the ability of one to take strategic action to achieve their self-defined goals. We use the respondent’s input in overall productive decisions, input in productive decisions relating to livestock, ownership of land and other assets, access to and input on decisions concerning credit, control over use of income in the household, work-life balance, and ability to visit important locations outside the home. In Table 9 below, we note that the percentage of women reported to have achieved adequacy in terms of input in productive decisions within the households, including livestock farming activities, was significantly greater compared to men (*p*-value < 0.001). When it comes to work-life balance (*p* < 0.001) and ability to visit important locations outside of the homestead (*p* = 0.035), we find a greater proportion of men doing better compared to the women. It is worth noting that productive decisions and productive decisions related to livestock production were mostly measured in terms of women’s labor contributions to productive activities. Female only households were also significantly limited when it came to visiting important locations.

Collective agency describes the ability to be a part of and/or mobilize people around common or shared concerns. This is captured by the group membership indicators. We find (Table 10) that a significantly greater percentage of women compared to men are members of both general community groups (*p*-value < 0.001) and influential groups (*p*-value < 0.001). This showed that women were significantly more engaged in their community based social networks.

Overall, the results above show at least 80% of women achieve adequacy in 5 out the 13 indicators namely, (i) membership in community groups, (ii) access to credit products and services, (iii) decisions regarding asset ownership, (iv) attitudes against domestic violence, and (v) control over income use. When it comes to the men, at least 80% were adequate in (i) decisions regarding asset ownership (ii) access to credit products and services (iii) attitudes against domestic violence, and (iv) decisions regarding income use (Figure 4).

#### 3.2.3. WELI Score

The respondent’s overall empowerment score was computed as the weighted average of her/his adequacy scores in the 13 indicators (all weighted 1/13). If he/she was adequate in 9 out of the 13 indicators, then that respondent was classified as empowered. Conversely, if inadequate in 4 or more indicators, then that respondent was classified as disempowered. These individual level scores were then aggregated to construct WELI score which is the weighted mean of two sub-indices: the Three Domains of Empowerment Index (3DE), with a weight of 90 percent, and the Gender Parity Index (GPI), with a weight of 10 percent.

When constructing GPI, only households where two adults, the man and woman were interviewed, were considered. In total, we had 95 dual adult households. However, during the data cleaning process, we were able to correctly identify 78 households with complete data from two household members. The household ID recording error was the main reason for excluding the 17 households at this stage. The aggregate WELI score for women in this study is 0.81; (Table 11). This is a weighted average of the 0.79 3DE score of women and the 0.92 GPI score. We take notice of the fact that there was no observable difference between the percentage of empowered women (50.3%) and men (51.5%). Of those women and men who are not yet empowered, the mean adequacy score is 0.58 in both genders, therefore these women and men achieve adequacy in an average of 58% of the 13 indicators. The study found that approximately 60% of the households achieved gender parity. The average empowerment gap between women who do not achieve gender parity and men in their households is 18%.

#### 3.2.4. Contribution of Indicators to Disempowerment

To identify the main factors contributing to disempowerment of respondents in this study, the disempowerment index was decomposed by indicator. Figure 5 shows the disempowerment graph. The length/size of the bar shows the extent to which that indicator contributes to disempowerment. The longer/bigger the bar, the higher the contribution. For women, work balance contributes the most to disempowerment followed by visiting important locations, respect among household members, self-efficacy and autonomy in decision making and membership in influential groups.

The indicators that contribute to men’s disempowerment include membership in influential groups, input in productive decisions, work balance, self-efficacy, respect among household members, control over use of income, and visiting important locations.

### 3.3. Participants Knowledge and Access to Vaccination and Vaccine Information

#### 3.3.1. Respondents Access to Vaccine Information

When we look at the differences in the respondent’s opinion about their ability to access information regarding vaccinating goats and chicken, we find that only 8.1% of women and 7.8% of men have access to information regarding vaccinating goats for CCPP. Similarly only 8.5% of women and 12.2% of men have access to information regarding chicken vaccination for ND. Overall, only 16.0% have access to information about any of the two vaccines. There is no observable difference by gender (Table 12).

#### 3.3.2. Respondents’ Ability to Vaccinate Their Animals

The survey also sought to find out the respondents’ ability to vaccinate their goats, and chickens against CCPP and ND, respectively. The results (Table 13) show less than 20% of the farmers (women 10.3/men 19.50 had the ability to vaccinate their animals against CCPP. The numbers were much higher for ND (32.5 women/37.8 men) and yet still fewer than 40% of the population. A slightly lower proportion of women are able to administer the vaccines against CCPP, and ND. This difference was however not statistically significant, at a 5% level.

To understand whether empowered women feel that they can participate to a great extent in vaccinating their animals or if their perceived extent of involvement is inversely related or not at all, we fit simple generalized linear regression models with 3DE score as the dependent variable. There was no statistically significant relationship between perceived ability to access information regarding vaccination or the ability of a woman to vaccinate livestock against CCPP/ND and the empowerment score (Table 14).

#### 3.3.3. Relationship between Training and Empowerment

Further, we checked whether attending training about livestock health helps to empower women. Again, we see no statistically significant association between training attendance and empowerment score.

#### 3.3.4. Relationship between Empowerment and Loss of Animals to Diseases

We sought to explore whether a woman’s empowerment level was associated with the reported vaccination rate and number of CCPP, and ND related deaths. A simple log-binomial regression model was fitted with the vaccine rate as a binary response variable and 3DE score as the independent. We find no significant relationship between the rate of vaccination and a woman’s empowerment score at 5% level. Two simple negative binomial regression models were run using CCPP and ND related death counts as response variables, with the empowerment score as the explanatory variable. The choice of negative binomial regression over Poisson models was informed by the overdispersion in the response variables. The results show an inverse and statistically significant relationship between the reported CCPP deaths in the past 12 months and a woman’s empowerment score (Table 15). For each unit increase in a women’s empowerment score, the CCPP death rate would be expected to decrease by a factor of 8.0%. There was however no statistically significant association between ND death rate and a woman’s empowerment score.

#### 3.3.5. Respondents’ Knowledge and Overall Empowerment Score

Lastly, we explore the impact of the respondent’s knowledge about vaccines on their overall empowerment score. For the question about the respondent’s knowledge about animal health, the response was recorded to a binary score where 0 represents “Not at all/small extent” and 1 represents “Medium extent/High extent”. This recording criterion is consistent with the approach used when creating the WELI study indices, particularly on input in productive decisions. A multiple generalized linear model was fitted with a 3DE score as the response variable and the three indicators (Table 16) as the covariates. We found that a respondent’s knowledge of where to purchase vaccines against CCPP or ND is significantly associated with the empowerment level. Individuals who know where to purchase vaccines, compared to those who do not, are likely to have 8.0% greater empowerment score, holding other factors constant. The study found no significant association between empowerment score and being knowledgeable about animal health, or access to information regarding the vaccines CCPP or ND, at a 5% level.

## 4. Discussion

As the crucial role played by rural women in livestock production and management is recognized, the level to which women benefit personally, socially and economically from keeping livestock and the extent to which engagement in these activities empowers or disempowers them is key. If women smallholder livestock farmers are empowered, there is increased livestock productivity, improved livelihoods and well-being and, enhanced food security, which contributes to gender equality and agency, access and control over resources [9,16].

We used the WELI quantitative survey as a baseline analysis to capture the position of women smallholder farmers, the factors that are currently causing disempowerment, their current limitations, to estimate the extent and nature of their participation in livestock production and vaccine access, determine the impact of women’s participation in livestock management on their families’ welfare status, with the intention of using this baseline data to target specific interventions that can improve access to and adoption of livestock vaccines, and ensure that women’s capabilities (skills, education, health) and agency (participation, voices, influence) are strengthened and recognized.

### 4.1. Household Structure, Age and Number of Household Members Influence Adequacy

Our findings reveal that it is important when analyzing gender factors with women smallholder farmers to consider other demographic factors, such as whether they belong to a female HH or Dual HH, age, and the number of people within a household. A higher percentage of women in female HH attained adequacy in indicators that were significant such as autonomy in income use, respect among household members, input in livestock productive decisions and work balance as compared to women in Dual HH. Female HH had more control over the income and decision making in their homes and were more respected because they controlled the financial resources. As the head of the household they are usually responsible for all or most of the household expenses or deciding how to spend the household income. We also presume that as HH, even though the workload is still exceptionally high, they can control how they spend the time and therefore more of them have a higher work balance. In most of the Dual HH, women have to consult their spouses or other males within the household and are limited in decision making. Johnston et al. reports that within their households, women in general play a limited role in household decision making and have little say in how household income is used despite undertaking most of the household work [35]. Although both men and women are affected by time constraints, women may experience more severe time trade-offs because of their heavier burden of unpaid work, and because their paid and unpaid work is often undertaken simultaneously, whereas men are seen as being more able to perform their activities sequentially [36].

Dungumaro (2007) argues that although a great body of literature suggests that female headed households are more deprived economically than their dual counterparts, gender analysis is important in order to have a better understanding of how female HH compare to Dual HH [37]. Results of such analysis provide information to support or negate some prevailing perceptions regarding female headed households. In reverse, a higher percentage of women in Dual HH attained adequacy in indicators related to visiting important locations and group participation. This aligns with most literature and focus group discussions because the workload for female HH is too high, limiting the time they have to participate in social activities, such as social help groups and network formation or visiting locations outside their villages. Studies have shown that it is especially difficult for female HH to participate in meetings, attend training and contribute to development activities, or even access financial credit. Sometimes this is a result of cultural stigmatization where female HHs are marginalized as a result of their status.

Age is also significant because older people including older women tend to have more autonomy and more control over decision making and resources than younger women. Older women are also more respected in communities, and sometimes gender relations are heavily influenced by community norms and values and community can be a major predictor of women’s empowerment than individual traits [38]. Our analysis showed that there exists a statistically significant relationship between the respondent’s age and WELI scores, after adjusting for gender. For each year’s increase in the respondent’s age, the 3DE score increased by a factor of 0.3%. Several studies found that as women get older they are more able to make decisions independently of men, and they gain more bargaining power [39]. Reviewing a number of such studies, Mason and Smith 2003 concluded that “*older women are argued to have more independence and empowerment than younger women because they have more experience with life, a better understanding of how to get what they want or need, a closer relationship with the husband, or because they have fulfilled certain social obligations to the husband and his family (for example, bearing children or sons) and thus are more trusted than are young wives, over whom tighter controls are maintained*” [38]. It is notable in our study that there is an inverse and statistically significant relationship between the household size and WELI score. For each additional member in the household, an individual’s expected 3DE score decreases by a factor of 1.9%, holding other factors constant. Households with more occupants therefore had a lower WELI score. Whereas other papers contend that fewer household members are directly proportional to women’s contribution to household income, in turn, reduced expenditure per capita in the household, pushing a significant number of families into poverty and preventing the escape of a significant number from poverty [40], our results aligns with papers that argue that an increase in household numbers reduces the economic prospects of families and societies. More people, especially of a younger age, means an increase in expenditure on household items and other areas, such as school fees [41].

### 4.2. Women’s Contribution towards Livestock Productive Activities and their Impact on WELI Score

In our study, we found that women contributed the most to agricultural and livestock productive activities and attained adequacy in this area compared to men. There was a significant difference between men and women in terms of all livestock production activities. On a deeper analysis of women’s contribution to livestock productive decision making, we find that most of the decisions are related to women’s roles and daily labor intensive livestock activities. The indicator is formulated in a way that the majority of it measures what women do/and their input in productive work. Women contributed the most to decisions related to feeding, milking, cleaning, breeding and slaughtering animals, checking animal health, providing disease prevention measures, marketing of animals and products, selecting breeding species and distributing the livestock workload within the household, and sharing livestock workload among household members. Only one question was related to whether women controlled any resources that came from livestock production. Our results aligned with published data that shows that women perform most of the productive work related to livestock. However, this indicator does not equal empowerment. In fact most publications argue that women perform most of the work and yet do not enjoy the benefits and control the resources that come from that and most of their unpaid work is not recognized. According to Aklilu et al. (2007), women perform all the day to day activities related to caring, feeding, cleaning, health and production of livestock [42]. A study by Odongo (2015) reveals that women have a key role in smallholder farm production [43]. Njuki and Sanginga (2013) report that despite their role in livestock production, women’s control has traditionally declined when productivity increases, and they do not access the resources and livestock products [9]. In addition women provide other types of unpaid labor, such as food preparation, feeding of young children, breastfeeding, child social stimulation and monitoring, collecting and/or treating water, collecting cooking fuel, managing household (and children’s) hygiene. It is worth noting that in this WELI analysis, engagement in livestock productive decision making has a major impact on the WELI score. There is a positive and statistically significant relationship between the number of livestock productive activities and the WELI score, at a 5% significance level. Holding other factors constant, for each unit increase in the number of livestock activities an individual is involved in, the expected WELI score increases by a factor of 2.0%, leading us to conclude that participation in livestock activities is empowering for women. However this conclusion is misleading because the indicator measured mostly women’s labor contribution to livestock production and did not consider control over livestock assets and resources, patterns of power and decision making over these resources, ability of women to decide, influence, control, and enforce any actions related to livestock, which should be the key aspects that contribute to empowerment. For livestock production, measurement of empowerment should not be limited to the labor women provide for activities. Women are known to invest much of their time in livestock rearing but rarely in activities that bring revenues [14]. The ability of a woman to take control over her own circumstances and to determine the path she wishes to take for the realization of her goals and her own self-worth in life is a measure of her empowerment, according to various authors [21,22,23]. Women’s empowerment enhances their ability to improve their decision making, attain instrumental outcomes, such as improved household well-being: (increased productivity, food security and livelihoods) [17,18] as well as their greater control over their resources and their bodies [19].

### 4.3. Contribution of Indicators to Disempowerment

In our study, although similar numbers of men and women (~50%) did not achieve empowerment, the indicators that contributed the most to disempowerment were different for men and women. Work balance was the most significant contributor to women’s disempowerment, followed by respect among household members, self-efficacy, autonomy in income use and visiting important locations. Work balance was couched in excessive workloads language which limited women’s ability to do many other things, including attending group meetings or earning income. The discussions of visiting important locations showed the extent of restrictions on women’s ability to leave the homestead owing to gender norms and lack of time, as well as the importance of mobility to enable women to attend group meetings and earn income.

It is worth noting that disempowered women were inadequate in three out of four of the indicators under intrinsic agency, ‘power within’ that is the process by which one develops a critical consciousness of their own aspirations, capabilities, and rights. These three were autonomy in income use, self-efficacy and respect among household members. Self-efficacy which referred to the ability/belief/capacity or confidence of women to control their behaviors, activities and motivations was an area identified by disempowered women. Many felt helpless to change their trajectory based on what they had lived through and relied on external help from their husbands for prosperity. Women described intrahousehold harmony in relation to respect among household members and felt it was important to them, both for its intrinsic value and because harmonious relations with husbands and in-laws would enable women to do more, including having greater capacity to move freely, attend group meetings, and earn income. For disempowered men, membership in influential groups was the most disempowering indicator, followed by self-efficacy, input in livestock productive decisions, and work balance. Men’s participation in influential groups was identified as low in the community because many men felt that many of the groups belonged to and were organized around women self-help groups. In Machakos, women joined local farmer self-help groups (SHGs) where they felt they had more control over the ownership of livestock and were using these as a safety net to protect against men taking over. These SHGs provided some basic credit and savings for purchase of goats and chicken, and other household items, and were formed as a means of collateral for credit access. The women observed that men had higher chances of accessing income or credit since they owned assets, such as land, that could be used as collateral. Women were also able to access livestock management training programs faster if they were within groups. The women also used the groups as a social support system. Input in livestock productive decisions was low because as discussed earlier most labor for livestock activities was provided by women. Women had a high adequacy in relation to group membership, which most of them referred to as self-help groups. They felt that group memberships were empowering to them providing opportunities for them to access training, information, resources and connections with others. It is worth noting that most women classified their self-help groups as influential groups to the community.

### 4.4. WELI SCORE as a Measure of Empowerment

The aggregate WELI score for women in this study is 0.81; this is a weighted average of the 0.79 3DE score of women and 0.92 GPI score. We take notice of the fact that there was no observable difference between the percentage of empowered women (50.3%) and men (51.5%). The 3DE score for men is 0.80, barely different from the women’s 0.79. The 3DE score represents the achievements of women in the sample across the 12 indicators of empowerment in WELI. The 1-3DE considers the number of women who are disempowered and the intensity of their disempowerment, or the number of indicators in which these women are not adequately empowered. This WELI score of 0.81 would make us assume that women in Machakos are very empowered. The closer to 1 the more empowered the women are. However, this data does not correlate with qualitative data collected through focus group discussions, key informant interviews and stakeholder engagements. Through focus groups we had discussions that characterized what the community members considered characteristics of empowerment- these are in Table 3 and included various statements such as—Can make decisions and take action to improve their livelihoods and incomes—Make decisions that enable them to participate and benefit from formal livestock markets—Can own successful businesses such as agrovets, distributors—Have more education a-college level and can therefore get better jobs and make more money—Increased ability to access credit as individuals or as group networks. In our FGDs the majority of the community members are not empowered and described their current limitations, including factors such as having limited access to vaccines, being economically disadvantaged, lacking decision making power and having limited resources, experiencing cultural limitations and gender bias,—lacking agency and decision making, lacking access to information on vaccines, and lacking access to credit facilities. We argue that the aggregate WELI score is not a correct reflection of this communities’ empowerment status and is therefore not a good measure of empowerment in this community. The WELI based on the Pro-WEAI rely heavily on instrumental agency indicators, consisting mainly of decision-making questions. These questions are mainly related to production, assets and credit, as indicated in Table 5. We have argued above that the in-put in livestock productive decisions indicator is not a true reflection of decision making because it focuses solely on women’s labor contribution to livestock production. The other indicators, such as access to land and other assets, do not consider the size of the land or the productivity of the land. In Machakos County, most families own ancestral land and yet these pieces of land are not productive enough to meet the family’s needs. Machakos is located in the semi-arid part of Kenya and most of the land is unusable. Just owning land in itself is not empowering. Similarly many women are members of self-help groups and can obtain credit but many times this credit is limited to what the women can contribute for each other, usually less than 10 dollars and they are still not able. These are issues that can apply to most rural communities in Africa. At the same time, for our communities in Kola and Kalama, men and women seem to be in the same status. The men are financially under-resourced and as disempowered as the women. Additionally, the reliance on instrumental agency implies that households with only female decision-makers are more likely to be identified as empowered by default, which is a known limitation of WEAI [32]. There is a need to examine the indicator contributions to empowerment to avoid these discrepancies.

### 4.5. Women’s Empowerment and Access to Livestock Services and Vaccination

Valuable descriptive data was collected regarding the study population and their self-reported practices around both ND and CCPP in this study using the additional vaccine module. The majority of respondents (approx. 90%) were not found to vaccinate against ND and felt they had only a small amount of knowledge regarding their livestock. Respondents overwhelmingly did not have access to cold-chain storage (a prerequisite for the use of the ND vaccine) and rarely, if ever, reported going to a veterinarian. These results are supported by recently collected data showing low ND vaccine adoption rates among small scale poultry farmers in Kenya [5]. Regarding CCPP, a larger proportion of individuals reported not vaccinating against the disease which is to be expected as Kenyan government officials control the CCPP vaccine and its distribution [6]. Goat livestock keepers noted similar rates of goat knowledge, cold-chain access and trips to the veterinarian as chicken keepers. Less animals were reportedly lost to CCPP than ND which is supported by the lower prevalence of CCPP versus ND.

Although no significant relationship was found between gender and animals lost to disease these results must be interpreted in light of the strengths and weaknesses of using the WELI survey for this originally unintended purpose. The WELI survey is derived from its well established Pro-WEAI predecessor and as a result the questions chosen to be included in this analysis and their phrasing was determined by experts in the field of agricultural empowerment [14]. The breadth and depth of its questions provide an extreme wealth of information. Despite the potential of using the WELI survey outside of its original intent, a number of weaknesses did limit its use for this specific analysis. Data regarding the number of animals lost to ND and CCPP, for example, was not collected in relation to herd size because there were no questions to assess that metric. This makes the number of animals lost to both diseases more difficult to interpret since data has shown that women tend to own more goats and poultry than their male counterparts. In general, the individual modules were also not designed to be compared to each other in this fashion and as a result the sample size of the analysis was reduced almost by half when the missing values in different modules were dropping after not overlapping. Finally, the sampling strategy of this WELI data collection was not designed to be representative of the source population but rather to represent a subset to be revisited later to monitor change in WELI scores after interventions were deployed. For this analysis, a representative sample would have been more ideal to enable generalization of the results. The lack of a clear consensus between this analysis and related existing literature regarding the relationship between gender and livestock lost to ND and CCPP stresses the as of yet unknown nature of this relationship and the need for further, more extensive study. Further studies to explore this relationship should focus on gaining a representative sample and designing a directed survey with questions designed to quantify the number of animals lost to ND and CCPP and an individual’s gender. Adopting a mixed methods parallel study design with both survey and focus group methodologies may also help to create a more accurate picture of this relationship.

### 4.6. Limitations of the Study

The WELI analysis did not include a vaccine module. The analysis of the additional vaccine module had to be done separately. Since the WELI is very long (takes almost three hours to administer), we had to be very concise about additional questions we could ask respondents to ensure we do not keep them for a very long time. There may have been other confounding factors that affected empowerment, such as landownership, geographical location, and socioeconomic status, that apply to this particular setting; although the study site is a good model for other locations. Our study did not include other known indicators/factors previously reported in other studies that influence empowerment, for example education and socio economic status at household level (respondent and spouse income levels), mainly because the WELI was a standardized tool with specific indicators, questions and these could not be changed. The criteria used to classify the respondents as either empowered or not, may not be applicable across different settings.

## 5. Conclusions

This paper uses the WELI with an additional vaccine module to analyze women smallholder livestock farmers’ empowerment, agency and access to livestock vaccines in Machakos district of Kenya. Women’s empowerment is mapped to 3 domains (3DE): intrinsic agency (power within), instrumental agency (power to), and collective agency (power with) measured against adequacy in 13 indicators. We conclude that the overall WELI score is very high (0.81/1) and is not an accurate indicator of smallholder women farmers empowerment in Machakos County. We argue that the results are skewed based on the framing of the livestock production roles, the structure of the households, as well as other confounding factors, such as the nature of the site, thereby presenting women smallholder farmers in Machakos County, as empowered in contrast to qualitative studies done in the same community that indicate otherwise. The most significant contribution to the WELI is women’s input in productive and livestock productive activities, which on a deeper analysis shows that most of the decisions are related to women’s roles and daily labor intensive livestock activities. The indicator is formulated in a way that the majority of it measures what women do/and their input in productive work. For livestock production, measurement of empowerment should not be limited to the labor women provide for activities. The WELI based on the Pro-WEAI relies heavily on instrumental agency indicators, consisting mainly of decision-making questions. These questions are mainly related to production, assets and credit. In addition, the reliance on instrumental agency implies that households with only female decision-makers are more likely to be identified as empowered by default. We suggest caution in using the WELI arbitrarily to measure women’s empowerment in the livestock sector, without analyzing individual characteristics of each community. However, the decomposability of the index allows us to disaggregate the drivers of change and examine how individual indicators are contributing to women’s disempowerment. This has allowed us to generate baseline data on the position and limitations of women smallholder farmers, and the factors that are currently causing disempowerment, and we can use these to target specific interventions that can improve women’s access to and adoption of livestock vaccines and their agency. Work balance was the most significant contributor to women’s disempowerment, followed by respect among household members, self-efficacy, autonomy in income use and visiting important locations. These factors need to be weighed heavily when formulating transformative projects for rural women because many development projects end up increasing the workload of women without necessarily empowering them. Our results indicate that the household structure (female headed or dual headed household), age of respondents and number of members in a household influence the adequacy score. Women smallholder livestock farmers report low CCPP and ND vaccination rates, minimal knowledge on livestock diseases, a lack of access to cold chain storage and rare visits to veterinarians; projects that specifically address these gaps would improve livestock productivity and enhance women’s influence and opportunities.

## Figures and Tables

**Figure 1 vaccines-10-01868-f001:**
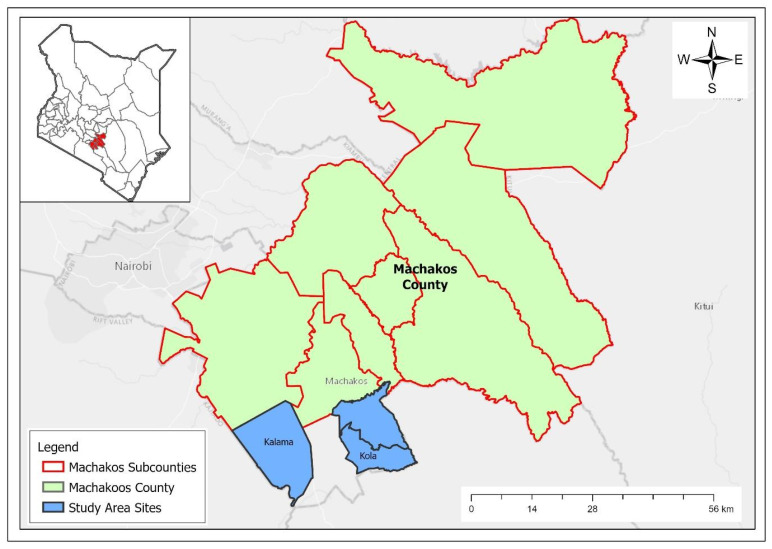
Map of Machakos County and Sub-Counties in Kenya. The wards of Kola and Kalama within Machakos Town Sub-county were selected for inclusion in this study.

**Figure 2 vaccines-10-01868-f002:**
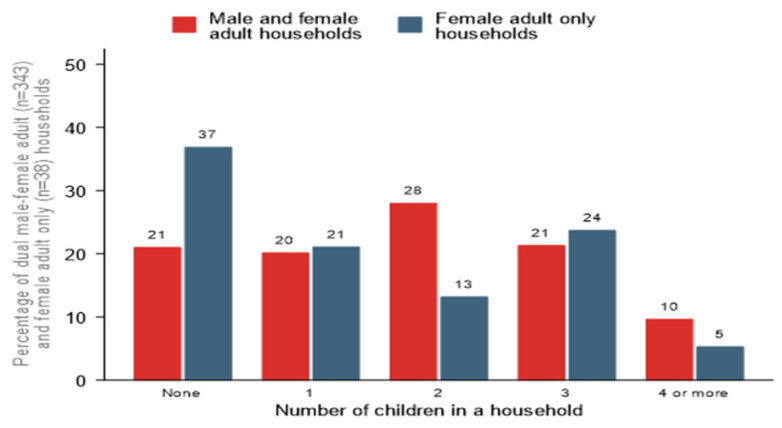
Number of children living in the surveyed households by household type.

**Figure 3 vaccines-10-01868-f003:**
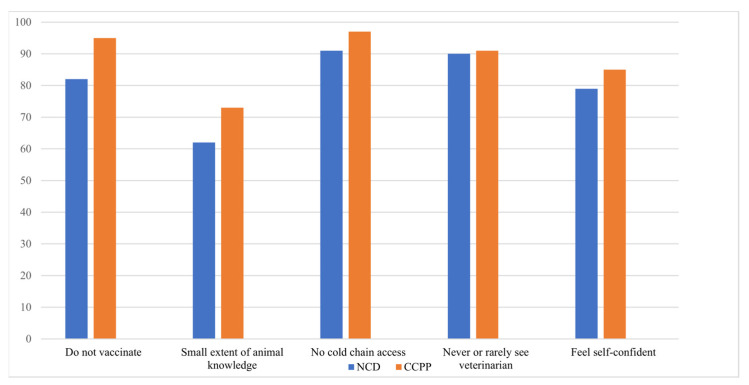
Perceptions of participants regarding behaviors and knowledge surrounding ND and CCPP.

**Figure 4 vaccines-10-01868-f004:**
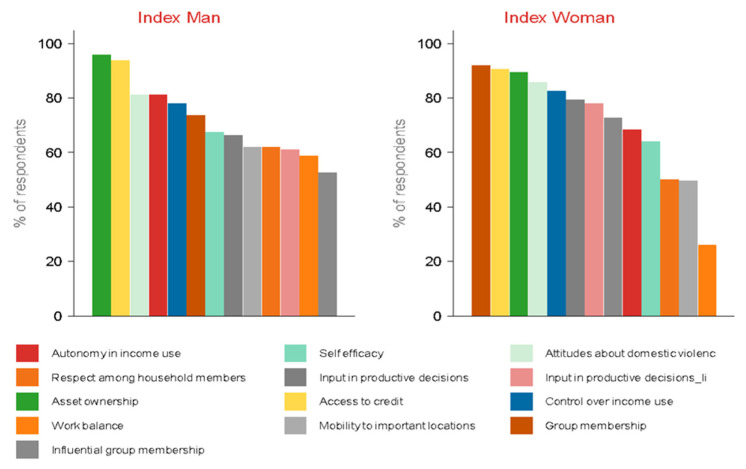
Share of respondent achieving adequacy per indicator by gender.

**Figure 5 vaccines-10-01868-f005:**
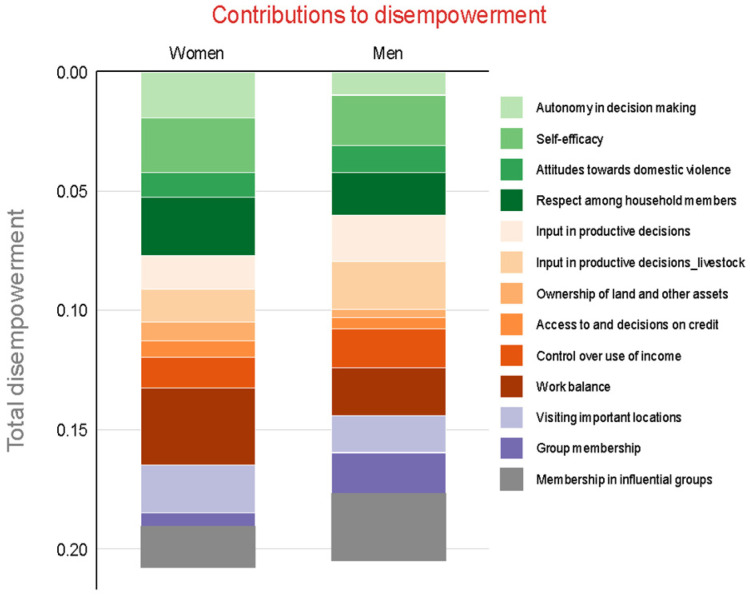
Contribution of each indicator to disempowerment by gender.

**Table 1 vaccines-10-01868-t001:** Tools used for data collection and the number of men and women who participated in each activity in Kola and Kalama wards of Machakos Town Sub-county, Kenya, 2019.

Tools Used	Nº of Events	Nº of Participants		
		Male	Female	Total Nº of People
WELI quantitative survey	1	95	285	380
Stakeholders’ meetings (SM)	3	22	14	36
Focus groups discussions (FGD)	10	4 (37)	6 (67)	104

**Table 2 vaccines-10-01868-t002:** Additional Livestock Vaccine module including the research questions and statistical tests done.

Independent Variable	Dependent Variable	Research Question	Hypothesis	Statistical Test
Demographic variable	Choose to vaccinate (categorical, binary)	Is there an association between vaccination rates among female livestock keepers and level of empowerment within the livestock sector?	Do empowered women vaccinate more?	Logistic regression
Knowing where to purchase vaccines (categorical, binary) Knowledge about animal health (categorical, ordinal) Access to information on vaccinating (categorical, ordinal)	WELI score (continuous)	Is there an association between vaccination knowledge among female livestock keepers and level of empowerment within the livestock sector?	Does knowledge about vaccines contribute to empowerment?	Multiple linear regression
WELI score (continuous)	Animals lost to disease (continuous)	Is there an association between animals lost to disease among female livestock keepers and level of empowerment within the livestock sector?	If women are more empowered do they lose less animals?	Linear regression
Attending training sessions about animals (continuous, binary)	WELI score (continuous)	Is there an association between attending training sessions for animals among female livestock keepers and level of empowerment within the livestock sector?	Is training helping to empower women?	Is training helping to empower women?

**Table 3 vaccines-10-01868-t003:** The socio-demographic data of participants.

	Male (Dual Adult Male Female HHs)	Female (Dual Adult Male Female HHs)	Female (Adult Female Only)	All Female
Mean age (SD)	57 (15)	48 (13)	56 (14)	49 (13)
Median age (Years)	59	47	57	48
Interquartile range	44–68	39–58	44–65	40–59
Min–Max age	25–90	22–79	22–85	22–85
No. of observations	95	247 ^1^	38	285

^1^ One index woman’s age data was missing.

**Table 4 vaccines-10-01868-t004:** Relationship between WELI score and demographic characteristics.

Covariate	Exp (b)	*p*-Value	[95% CI]
Respondent’s gender (Ref group = Index man)	1.020	0.459	0.967–1.076
Respondent’s age in years	1.003	<0.001	1.002–1.005
Household size	0.981	0.005	0.967–0.994

Number of observations (*n* = 380).

**Table 5 vaccines-10-01868-t005:** Share of respondents (%) participating in important livestock activities.

Activity	Men	Women	*p*-Value
Animal feeding	58.9%	92.0%	<0.001
Checking animal health	68.4%	88.8%	<0.001
Disease preventative measures	57.7%	60.3%	0.744
Milking animals	66.7%	85.0% ^1^	-
Cleaning animals	38.1%	90.2%	<0.001
Slaughter animals	45.4%	64.3%	<0.001
Breeding	39.0%	65.2%	<0.001
Marketing of live animals and products from live animals	40.3%	59.4%	<0.001
Selecting which species and breeds to rear	28.6%	33.3%	0.689
Sharing livestock workload among household members	75.4%	96.0%	<0.001

^1^ No of respondents too few to calculate *p* value since animals of interest were goats—they are rarely milked.

**Table 6 vaccines-10-01868-t006:** Relationship between empowerment level and participation in agricultural, livestock activities.

Covariate	Exp (b)	Std. Error	[95% CI]	*p*-Value
Number of agricultural activities in which individual participates in	1.07	0.01	1.04–1.10	<0.001
Number of livestock activities in which individual participates in	1.02	0.01	1.01–1.03	0.001
Gender of the respondent (Ref group = Male)	0.98	0.03	0.93–1.03	0.457
Respondent age	1.00	<0.01	0.99–1.00	0.068
Household size	0.98	0.01	0.97–0.99	0.001

No. of observations = 377.

**Table 7 vaccines-10-01868-t007:** Characteristics of empowerment as described by participants.

Economic Independence	Knowledge and Skills	Opportunities for Networking	Autonomy in Decision Making
Can negotiate for better buying prices of poultry and goats.	Access to informal training on vaccine handling and administration	Organized into networks/with activities that empower fellow women on VVC and livestock production	Can make decisions and take action to improve their livelihoods and incomes
Can increase the size of their flocks-Own more goats and chicken that they can sell for more profit	Access to information on animal health and entrepreneurship so they can improve their businesses	Increased ability to access credit as individuals or as group networks	Make decisions that enable them to participate and benefit from formal livestock markets
Can own successful businesses such as agrovets, distributors	They are engaging their partners, family members and sharing information and skills obtained from trainings	Their groups can influence government decisions and structures	Can move vertically upwards into influential positions e.g., animal health assistants, vets, directors
Can have access to credit and resources similar to the men	Have more education at-college level and can therefore get better jobs and make more money	Their networks have more savings, which they can rely on in hard times	Can be recognized by their husbands and community leaders as contributors

**Table 8 vaccines-10-01868-t008:** Share of respondents attaining adequacy in terms of intrinsic agency.

	Male Respondents Dual Adult HHs (%)	Female Respondents Dual Adult HHs and Female Only HHs	*p*-Value	Female Respondents	
				Adult Female Only HHs (%)	Dual Adult HHs (%)
Autonomy in income use	81.1	68.5	0.019	79.0	66.9
Self-efficacy	67.4	64.0	0.550	60.5	64.5
Attitudes about domestic violence	81.1	85.7	0.282	86.8	85.5
Respect among household members	62.1	50.0	0.041	65.8	47.6
No of observations	95	286		38	248

**Table 9 vaccines-10-01868-t009:** Share of respondents attaining adequacy in terms of instrumental agency.

	Male Respondents (%)	Female Respondents (%)	*p*-Value	Female Respondents	
				Adult Female Only HHs (%)	Dual Adult HHs (%)
Input in productive decisions	66.3	86.7	<0.010	86.8	86.7
Input in productive decisions–livestock	61.1	92.3	<0.001	94.7	91.9
Ownership of land other assets	95.8	89.5	0.063	92.1	89.1
Access to and decisions on credit	93.7	94.1	0.895	92.1	94.4
Control over use of income	78.0	82.5	0.316	79.0	83.1
Work balance	59.0	26.2	<0.001	31.6	25.4
Visiting important locations	62.1	49.7	0.035	39.5	51.2
No. of observations	95	286		38	248

**Table 10 vaccines-10-01868-t010:** Share of respondents attaining adequacy in terms of collective agency.

	Male Respondents (%)	Female Respondents (%)	Chi-sq. (*p*-Value)	Adult Female Only HHs (%)	Female in Dual Adult HHs (%)
Group membership	73.7	92.0	<0.001	89.5	92.3
Influential Group membership	52.6	72.7	<0.001	68.4	73.4
No of observations	95	286		38	248

**Table 11 vaccines-10-01868-t011:** WELI score results.

Indicator	Men	Women
Number of observations	95	286
3DE score	0.80	0.79
Disempowerment score (1-3DE)	0.20	0.21
% achieving empowerment	51.5%	50.3%
% not achieving empowerment	48.5%	49.7%
Mean 3DE score for not yet empowered	0.58	0.58
Mean disempowerment score (1-3DE)	0.42	0.42
Gender Parity Index (GPI)		0.92
Number of dual-adult households		78
% achieving gender parity		57.7%
% not achieving gender parity		42.3%
Average empowerment gap		0.18
WELI score		0.81

**Table 12 vaccines-10-01868-t012:** Share of respondents (%) with access to information about vaccination by gender.

Access to Information about Vaccination	Men	Women	Chi-sq. (*p*-Value)
Have access to information regarding vaccinating goats for CCPP	7.8%	8.1%	0.008 (0.927)
Have access to information regarding vaccinating chicken for NCD	12.2%	8.5%	1.127 (0.288)
Have access to information regarding any of the two vaccinations	18.9%	16.0%	0.339 (0.560)

**Table 13 vaccines-10-01868-t013:** Share of respondents (%) able to vaccinate livestock against CCPP and ND by gender.

Ability to Vaccinate	Men	Women	Chi-sq. (*p*-Value)
Farmer is able to vaccinate goats against CCPP	19.5%	10.3%	4.650 (0.098)
Farmer is able to vaccinate chicken against NCD	37.8%	32.5%	0.928 (0.629)
Able to vaccinate livestock against any of the two diseases	46.9%	39.3%	1.438 (0.230)

**Table 14 vaccines-10-01868-t014:** Relationship between vaccine information access, administration, and empowerment level.

Independent Variables	Exp (b)	Std. Error	[95% CI]	*p*-Value
Perceived ability to access information regarding vaccination (Ref group = None/small extent). No. of observations = 231	0.94	0.04	0.87–1.02	0.146
Able to vaccinate livestock against CCPP or NCD (Ref group = No). No. of observations = 244	0.95	0.03	0.90–1.00	0.066
Attended training about goat or chicken health in the past 12 months (Ref group = No). No. of observations = 22	0.91	0.24	0.54–1.54	0.727

**Table 15 vaccines-10-01868-t015:** Relationship between empowerment levels, vaccination rates and livestock death counts.

Dependent Variables	Model Used	Exp (b)	Std. Error	[95% CI]	*p*-Value
Vaccination rate No of obs. = 236	Log binomial model (Reporting Risk ratio)	0.94	0.70	0.22–4.07	0.938
Reported number of CCPP deaths in the past 12 months No of obs. = 232	Negative binomial regression (Reporting Incidence Rate ratio)	0.08	0.06	0.02–0.34	0.001
Reported number of NCD deaths in the past 12 months No of obs. = 270		1.35	0.52	0.64–2.88	0.431

**Table 16 vaccines-10-01868-t016:** Relationship between self-reported vaccination knowledge among female livestock keepers and level of empowerment.

Covariates	Exp (b)	Std. Error	[95% CI]	*p*-Value
Knows where to purchase vaccine against CCPP or NCD (Ref group = No).	1.08	0.03	1.02–1.15	0.007
Knowledgeable about animal health, goat or chicken (Ref group = Not at all/small extent)	1.01	0.04	0.94–1.08	0.791
Have access to information regarding any of the two vaccinations (Ref group = No)	0.93	0.04	0.85–1.01	0.099

No. of observations = 301.

## Data Availability

All relevant data are within the manuscript.

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
