# Peer review of "Using the Women Empowerment in Livestock Index (WELI) to Examine Linkages between Women Smallholder Livestock Farmers’ Empowerment and Access to Livestock Vaccines in Machakos County of Kenya: Insights and Critiques"

_vaccines, 2022, doi:10.3390/vaccines10111868_

Round 1

Reviewer 1 Report

This is a very interesting piece of work examining the relationship of women's empowerment and their access to livestock vaccines in an area where people depends on the livestock farming to make a living. A systematic investigation with 400 participants was carried out. Although the research concludes that the WELI score is not an accurate indicator of women's empowerment in the area, the decomposability of the index allowed some insightful analysis. The paper can be published subject to the improvement of the following area:

1. In the introduction part, a more comprehensive literature review on general women empowerment and measuring index should be carried out, covering its application to other industries.  At least the following two references are recommended to be added ( the first one is the classic piece of work on women empowerment index, and the 2nd one is a most recent application of empowerment implementation in another industry):

(1) Lombardini, S.; Bowman, K.; Garwood, R. A ‘How to’ Guide to Measuring Women’s Empowerment: Sharing Experience from Oxfam’s Impact Evaluation. Oxfam GB. 2017. Available online:https://www.oxfam.org.au/wp-content/uploads/2018/02/A-how-to-guide-to-measuring-womens-empowerment.pdf

(2) Wang, C.C.; Mussi, E.; Sunindijo, R.Y. Analysing Gender Issues in the Australian Construction Industry through the Lens of Empowerment. Buildings 202111, 553. https://doi.org/10.3390/buildings11110553

2. The authors state that the weights for the two subindexes 3DE and GPI being 90% and 10% respectively are arbitrary but reflects the emphasis on 3DE... etc. This is not satisfactory in a scientific research, the determination of the weightage needs to be better justified.

3. The abstract can be shortened - just highlight the most important findings.

Author Response

Response to Reviewer 1 Comments

Point 1: In the introduction part, a more comprehensive literature review on general women empowerment and measuring index should be carried out, covering its application to other industries.  At least the following two references are recommended to be added ( the first one is the classic piece of work on women empowerment index, and the 2nd one is a most recent application of empowerment implementation in another industry):

(1) Lombardini, S.; Bowman, K.; Garwood, R. A ‘How to’ Guide to Measuring Women’s Empowerment: Sharing Experience from Oxfam’s Impact Evaluation. Oxfam GB. 2017. Available online:https://www.oxfam.org.au/wp-content/uploads/2018/02/A-how-to-guide-to-measuring-womens-empowerment.pdf

(2) Wang, C.C.; Mussi, E.; Sunindijo, R.Y. Analysing Gender Issues in the Australian Construction Industry through the Lens of Empowerment. Buildings 2021, 11, 553. https://doi.org/10.3390/buildings11110553

Response 1: We have modified the introduction to include more information on women empowerment, different empowerment indices currently in use, their focus and their limitations. We added lines 94-101 and lines 115-150. We are also grateful for the suggested papers and have integrated information from them accordingly.

Point 2: The authors state that the weights for the two subindexes 3DE and GPI being 90% and 10% respectively are arbitrary but reflects the emphasis on 3DE... etc. This is not satisfactory in a scientific research, the determination of the weightage needs to be better justified.

Response 2: We have deleted this sentence and added a sentence from the original developers of the 3DE and Gender Parity Index; lines 276-277.

Point 3: The abstract can be shortened - just highlight the most important findings.

Response 2: We have shortened the abstract.

Reviewer 2 Report

Well done study and presentation of the findings.  Although the manuscript is very lengthy, I do not think it can be reduced without losing its intention and aims.    I have two general suggestions that the authors may considered:

1.       The appropriate application of the tools for this study are sensitive to culture, geographical location, and other social factors.  Therefore, the authors may indicate the limitation of their findings to this specific study site, although the study site is good model for other locations.

2.       The submission can be improved if a set of recommendations, including the limitations, is addressed.

I wish the authors great successful endeavors to pursue this important topic.   

Author Response

Response to Reviewer 2 Comments

Point 1: The appropriate application of the tools for this study are sensitive to culture, geographical location, and other social factors.  Therefore, the authors may indicate the limitation of their findings to this specific study site, although the study site is good model for other locations.

Response 1: We have added limitations to study to include geographical location, socioeconomic status of both men and women, and landownership; lines 865-867. 

Point 2: The submission can be improved if a set of recommendations, including the limitations, is addressed.

Response 2: We have added specific location in the conclusion to emphasize that this applies to the study location; line 883.